# Reinforcement Learning for Mean-Field Game

**Mridul Agarwal** [1,*]**, Vaneet Aggarwal** [1,2,*]**, Arnob Ghosh** [3] **and Nilay Tiwari** [4]

1   School of Electrical and Computer Engineering, Purdue University, West Lafayette, IN 47907, USA
2   School of Industrial Engineering, Purdue University, West Lafayette, IN 47907, USA
3   Department of Electrical and Computer Engineering, Ohio State University, Columbus, OH 43210, USA; arnob008@gmail.com
4   Department of Electrical Engineering, I.I.T. Kanpur, Kanpur 208016, UP, India; nilay47@gmail.com
*   Correspondence: agarw180@purdue.edu (M.A.); vaneet@purdue.edu (V.A.)

**Abstract:** Stochastic games provide a framework for interactions among multiple agents and enable a myriad of applications. In these games, agents decide on actions simultaneously. After taking an action, the state of every agent updates to the next state, and each agent receives a reward. However, finding an equilibrium (if exists) in this game is often difficult when the number of agents becomes large. This paper focuses on finding a mean-field equilibrium (MFE) in an action-coupled stochastic game setting in an episodic framework. It is assumed that an agent can approximate the impact of the other agents' by the empirical distribution of the mean of the actions. All agents know the action distribution and employ lower-myopic best response dynamics to choose the optimal oblivious strategy. This paper proposes a posterior sampling-based approach for reinforcement learning in the mean-field game, where each agent samples a transition probability from the previous transitions. We show that the policy and action distributions converge to the optimal oblivious strategy and the limiting distribution, respectively, which constitute an MFE.

**Keywords:** reinforcement learning; mean-field game; equilibrium

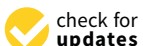



## 1. Introduction

### 1.1. Motivation

We live in a world where multiple agents interact repeatedly in a common environment. For example, multiple robots interact to achieve a specific goal. Multi-agent reinforcement learning (MARL) refers to the problem of learning and planning in a sequential decision-making system with unknown underlying system dynamics. The agents need to learn the system dynamics by trying different actions and observing rewards received over time. Learning in a MARL is fundamentally different from the traditional single-agent reinforcement learning (RL) problem since agents not only interact with the environment but also with each other. Thus, an agent, when trying to learn the underlying system dynamics, has to consider the action taken by the other agents. Changes in the policy (or actions) of any agent affect the others and vice versa.

One natural learning algorithm is to extend existing RL algorithms to the MARL by assuming that the other agents' actions are independent. However, studies show that a *smart* agent which learns the joint actions of the others performs better as compared to the agent that does not learn the joint action of other agents [1,2]. For any agent, the actions of other agents become a part of the state. This results in the state space increases exponentially as the number of agents increases. When the agents are strategic, i.e., they only want to take actions that maximize their utility (or value), Nash equilibrium is often employed as the equilibrium concept. The existing equilibrium solving approaches work for some restricted games when there exists an adversarial equilibrium or coordination equilibrium [3]. Also, these approaches can handle a handful of agents because of the exponential increase in the state space. The computational complexity of finding Nash

Equilibrium at every stage game prevents applications of these approaches in games where the number of agents is large [4].

In this paper, we consider MARL as an environment where a large number of agents co-exist. Similar to [5], we utilize a mean-field approach where we assume that the Q-function of an agent is affected by the mean actions of the others. Mean-field game drastically reduces the complexity, since an agent only needs to consider the empirical distribution of the actions played by other agents. Such mean-field games exist in several domains. For example, the mean-field game is observed in a cyber-security game where a large number of agents such as terminal nodes or servers make individual decisions about their security [6,7]. However, the ultimate security depends on the decisions made by other agents as well. For example, consider a network of computers, where there are a large number of agents and each agent manages a computer. If an agent invests heavily in building firewalls, its computer can still be breached if other agents' computers are not secure. In the security game, each agent invests a certain amount to attain a security level. However, the investment level depends on the investment made by the other agents. If the number of agents is large, the game can be modeled as the mean-field game as the average investment made per agent impacts the decision of an agent.

Another example of a mean-field game is the demand response price in the smart grid [8,9]. The utility company sets a price based on the average demand per household. Hence, if at a certain time the average demand is high, the utility company may increase the price. The agent now might want to reduce its own consumption to decrease its costs resulting from the increased price. Mean-field equilibrium is the equilibrium concept in the mean-field game.

### 1.2. Contribution

We seek to obtain a model-based RL algorithm to find the mean-field equilibrium in an episodic-set up. To the best of our knowledge, this is the first work on an episodic RL algorithm for mean-field equilibrium. We consider an *oblivious* strategy [10,11], where each agent takes an action based only on its state. Thus, even though the transition probability and reward depend on the empirical distribution of the agents' actions and states, an agent only seeks policy based on its own state. Hence, an agent does not need to track the policy state of the other agents.

In our algorithm, we maintain a history of the samples and sample an MDP that fits the history with high confidence at the start of the episode. The policy is then computed with this MDP. The agents then play the computed policy and collect samples over the steps of the episode. We show that such an algorithm converges to equilibrium.

### 1.3. Related Literature

Unlike the standard literature on the mean-field equilibrium on stochastic games [10,12,13], we consider that the transition probabilities are unknown to the agents. Instead, each agent learns the underlying transition probability matrix using a readily implementable posterior sampling approach [14,15]. All agents employ the best response dynamics to choose the best response strategy which maximizes the (discounted) payoff for the remaining episode length. We show the asymptotic convergence of the policy and the action distribution to the optimal oblivious strategy and the limiting action distribution respectively. We estimate the value function using backward induction and show that the value function estimates converge to the optimal value function of the true distribution. We also use the compactness of state and action space to show that the converged point constitutes a mean-field equilibrium (MFE).

Ref. [5] considers a variant of the Mean-field game where the state is the same across the agents. Unlike [5], we consider a generalized version of the game where the state can be different for different agents. Further, we do not consider a game where adversarial equilibrium or coordinated equilibrium is required to be present. We also do not need to track the action and the realized Q-value of other agents as was the case in [5].

Recently, the authors of [16] studied a policy-gradient based approach to achieve mean-field equilibrium. The authors of [17,18] considered variants of Q-learning to achieve Nash equilibrium. Actor-critic based algorithms have been analyzed in [19]. The authors of [20] consider a deep neural network-based algorithm for mean-field games. In contrast, this paper considers a posterior sampling-based approach. As noted in [21], model-based approaches converge faster than the model-free approaches in general. Thus, understanding of theoretical properties of the model-based approaches is an important problem. Further, all these papers require huge storage space as the policy depends on the actions of the other agents whereas in our setting we provide a policy that is oblivious of the states and actions of other agents.

The authors of [22,23] compute mean-field equilibrium for a setting where the evolution of the state of an agent does not depend on the action or states of the other agents. However, in our setting, we consider a generic setting where the both reward and the next state of an agent depend on the actions as well as the states of the other agents. Thus, though the mean-field game converges to the potential game in the above papers, in our setting, the game does not converge to the potential game. Hence, finding an equilibrium strategy is more challenging in our setting. The key contribution of the paper is a Posterior-sampling based algorithm that is used by each agent in a multi-agent setting, which is shown to converge to a mean-field equilibrium. The proposed algorithm does not assume the knowledge of transition probabilities and learns them using a posterior sampling approach. Further, our algorithm is computationally efficient as the policy only depends on the current state of an agent, i.e., it is oblivious of the states and actions of other agents.

## 2. Background

### 2.1. Multi-Player Stochastic Game

An $n$-player stochastic game is formalized by the system dynamics tuple $\mathcal{M}^* = \{\mathcal{S}, \mathcal{A}, P, r, \tau, \rho, \gamma\}$. The agents are indexed by the set $[n] = \{1, 2, \cdots, n\}$. The state of the $i$th agent at time $t$ is given by $s_{i,t} \in \mathcal{S}$, where $\mathcal{S}$ is the state space set. $\mathcal{A}(s)$ is defined as the set of the feasible actions any agent can take in state $s$. $\mathcal{A}$ is the action space set defined as $\bigcup_{s \in \mathcal{S}} \mathcal{A}(s)$. We also assume that both $\mathcal{S}$ and $\mathcal{A}$ are finite sets. Since finite sets are also compact, the assumption allows us to use results from previous works of [10]. Since we have $n$ agents in our system, the combined state space of the system becomes $\mathcal{S}^{\times n} = \mathcal{S} \times \mathcal{S} \times \cdots \times \mathcal{S}$, and the combined action space of the system becomes $\mathcal{A}^{\times n} = \mathcal{A} \times \mathcal{A} \times \cdots \times \mathcal{A}$. Let $\mathbf{s}_t = \mathbf{s} \in \mathcal{S}^{\times n}$ be a vector of length $n$, and the $i$th element of $\mathbf{s}$ denotes the states of the $i$th agent at time $t$. Similarly, let $\mathbf{a}_t = \mathbf{a} \in \mathcal{A}^{\times n}$ be a vector of length $n$, and the $i$th element of $\mathbf{s}$ denotes the action taken by the $i$th agent at time $t$.

If the agents play joint action $\mathbf{a}_t = \mathbf{a} \in \mathcal{A}^{\times n}$ the next state of the system $\mathbf{s}_{t+1} = \mathbf{s}' \in \mathcal{S}^{\times n}$ follows the probability distribution $P(\mathbf{s}_{t+1} = \mathbf{s}' | \mathbf{s}_t = \mathbf{s}, \mathbf{a}_t = \mathbf{a})$. Along with the state updates, $i$th agent also receives a reward $r_{i,t} = R_i(\mathbf{s}_t, \mathbf{a}_t, \mathbf{s}_{t+1}) \in [0, 1]$. We further assume that the reward function $R_i(\cdot)$ independent of the agent $i$, and each player is trying to optimize for the same reward. Hence, we can drop the subscript $i$ in the reward function $R_i$. However, since each agent can be in different state, or play different action, their individual rewards will be different and we still use the subscript $i$ to differentiate between the instantaneous reward of the $i$th agent. The constant $\gamma \in [0, 1)$ is the discount factor, and $\rho$ is the initial state distribution such that $\mathbf{s}_0 \sim \rho$.

We consider an episodic framework where the length of the time horizon or the length of episodes is $\tau$. State space set $\mathcal{S}$, action space set $\mathcal{A}$, $\tau$ are known and need not be learned by the agent. We consider that the game is played in episodes $k = 0, 1, 2, \cdots$. In each episode, the game is played in discrete steps, $j \in [\tau] = \{0, 1, \cdots, \tau - 1\}$. The episodes begin at times $t_k = k\tau, k = 0, 1, 2, \cdots$. At each time $t$, the state of the agent $i$ is given by $s_{i,t}$, the agent selects an action $a_{i,t}$, agent observes a scalar reward $r_{i,t}$ and the state transitions to the state $s_{t+1}$. Let $H_{i,t} = (s_{i,1}, a_{i,1}, r_{i,1}, \cdots, s_{i,t-1}, a_{i,t-1}, r_{i,t-1}, s_{i,t})$ denote the history available to the agent $i$ till time $t$.

### 2.2. Mean-Field Game

In a game with a large number of players, we might expect that the distribution of agents over the action space carries more meaning than the actions themselves. It is intuitive that a single agent has a negligible effect on the game as the number of agents increases. The effect of other agents on a single agent's payoff is only via the action distribution of the population. This intuition is formalized in the *mean-field game*. We now formally define the mean-field game and equilibrium concepts.

First, we define few notations. Let $\alpha_{-i,t}(a) : \mathcal{A} \to [0,1]$ be the fraction of the agents (excluding agent $i$) that take action $a \in \mathcal{A}$ at time $t$. Mathematically, we have

$$\alpha_{-i,t}(a) = \frac{1}{n-1} \sum_{m \in [n] \setminus \{i\}} \mathbb{1}(a_{m,t} = a), \tag{1}$$

where $\mathbb{1}(a_{j,t} = a)$ is the indicator function that the agent $j$ takes action $a$ at time $t$. Since we assume an episodic framework, $\alpha_{-i,t}(a)$ can be different at each time index in an episode and also across the episodes. The episodic nature of the problem will be used later (in (9)) to define convergence to a value that depends only the time index in the episode. Further, since each agent selects exactly one action from $\mathcal{A}$, we have

$$\alpha_{-i,t}(a) \geq 0 \ \forall \ a \in \mathcal{A}, \text{ and } \sum_{a \in \mathcal{A}} \alpha_{-i,t}(a) = 1$$

Similar to distribution of agents over actions, we define $f_{-i,t} : \mathcal{S} \to [0,1]$ as the distribution of agents (excluding agent $i$) over the state space $\mathcal{S}$.

$$f_{-i,t}(a) = \frac{1}{n-1} \sum_{m \in [n] \setminus \{i\}} \mathbb{1}(s_{m,t} = s), \tag{2}$$

In a mean-field game, every agent $i \in [n]$ assumes that its next state $s_{i,t+1}$ is randomly distributed according to the transition probability distribution $P_i$ conditioned on agent's current state $s_{i,t}$, the action taken $a_{i,t}$ and other agents' distribution over actions $\alpha_{-i,t}$. Also, the reward is function of the agents current state and action and the next state.

$$s_{i,t+1} \sim P_i(\cdot|s_{i,t}, a_{i,t}, \alpha_{-i,t}) \tag{3}$$

$$r_{i,t} = \phi(s_{i,t}, a_{i,t}, \alpha_{-i,t}, s_{i,t+1}) \tag{4}$$

Thus the agent does not need to concern itself with the actions of the other agents, as the population action distribution $\alpha_{-i,t}$ becomes a part of the environment. This updated environment dynamics can now be used in decision-making. Note that the distribution of the population action $\alpha_{-i,t}$ could be explicitly taken into account for deciding an action as well. Note that the reward may also depend on $f$ the state distribution of other agents. The analysis would have been similar.

**Example 1.** *We now provide an example drawn from a real-life application that simulates our setting. Suppose we consider the scenario of malware spreading. The state $X_{i,t} = 0$ means that an agent is vaccinated and can not infect at t. On the other hand, $X_{i,t}$ can vary between 0 and 1 with n quantization levels. The action space of an agent is $a_{i,t} = \{0, 1\}$. If $a_{i,t} = 0$ then the agent does not take any action. If $a_{i,t} = 1$, agent i takes action in order to protect itself. In order to simplify the model, we consider the following state evolution model*

$$X_{i,t+1} = \begin{cases} \min\{X_{i,t} + \omega_{i,t}\alpha_{-i,t}(0), 1\} & \text{if } a_{i,t} = 0 \\ 0 & \text{if } a_{i,t} = 1 \end{cases}$$

*Note that if all the other agents have taken action 1 i.e. they protect themselves, then an agent has less chance to be infected. Thus, the state is smaller when the fraction of agents taking action 0 (i.e., $\alpha_{-i,t}(0)$) is smaller. $\omega_{i,t}$ is a noise who has its value in $\{1/n, 2/n, \ldots, 1\}$.*

*The reward function for agent i is defined as follows $r_{i,t} = -X_{i,t}\alpha_{-i,t}(1) - \lambda a_{i,t}$. The reward function increases if the number of agents who have protected themselves is higher. $\lambda$ is the cost that depends on the action. This example explains the necessity to compute the mean-field equilibrium.*

Note that in general action of an agent should depend on the action distribution of other agents. However, Proposition 1 from [10] says that under equilibrium, an oblivious strategy performs as well as a strategy that considers other agents' actions. Thus, the strategy does not need to explicitly consider the value of $\alpha_{-i,t}$.

**Definition 1.** *An agent $i \in [n]$ is said to follow an oblivious deterministic strategy $\pi_i$ when the agent selects an action considering only time index j in an episode and current state $s_{i,j}$.*

$$\pi_i : \mathcal{S} \times [\tau] \to \mathcal{A} \tag{5}$$

$$a_{i,j} = \pi_i(s_{i,j}, j) \tag{6}$$

*For the rest of the paper, we will focus on oblivious deterministic strategy for all agents.*

### 2.3. Value Function, Q Function and Policy

We now define a value function for an agent $i \in [n]$ for oblivious policy $\pi_i$ at $l$th time step in an episode as:

$$V_{i,\pi_i,l}(s|\alpha_{-i,l}) = \mathbf{E}_{P_i,\pi_i}\left[\sum_{j=l}^{\tau-1} \gamma^{j-l} r_{i,t}|s_{i,l} = s\right] \tag{7}$$

$$= \mathbf{E}_{P_i,\pi_i}\left[\sum_{j=l}^{\tau-1} \gamma^{j-l} \phi(s_{i,j}, a_{i,j}, s_{i,j+1})|s_{i,l} = s\right]. \tag{8}$$

The expectation in Equation (8) is taken over the actions taken from time step $l$ and the states visited after time step $l$ in an episode. We will consider the rest of the definitions from some $i$th agent's perspective, $i \in [n]$, so subscripts $i$ and $-i$ will be dropped for brevity.

We note that the action space and state space are finite and hence the set of strategies available to the players is also finite. The player adopts the lower myopic best response dynamics to choose the policy. A lower myopic policy selects an action with the lowest index among the actions that maximize the value function. As time proceeds, the strategies and the action distribution converge to the asymptotic equilibrium [10].

Let $\alpha_j^* \in [0,1]^{|\mathcal{A}|}$ be the limiting population action distribution for $j$th time index in episode $k$. We note that due to the episodic framework, the limiting action distribution depends on the index in an episode. Then, from the definition of limit, for every $\epsilon > 0$ there exist a $K_\epsilon < \infty$ such that for all $k > K_\epsilon$, we have

$$||\alpha_{k\tau+j} - \alpha_j^*||_2 < \epsilon \tag{9}$$

where $||.||_2$ denotes the $\ell_2$ norm.

The value function defined in Equation (8) satisfies the Bellman-property for finite horizon MDPs, given by

$$V_{\pi,l}(s|\alpha_l^*) = \mathbf{E}_{P_i,\pi_i}\left[\sum_{j=l}^{\tau-1} \gamma^{j-l} r_{i,j}|s_{i,l} = s\right] \tag{10}$$

$$= \sum_{s'\in S} P_i(s'|s,a,\alpha_l^*)r_{i,l} + \mathbf{E}_{P_i,\pi_i}\left[\sum_{j=l+1}^{\tau-1} \gamma^{j-l} r_{i,j} = s'\right] \tag{11}$$

$$= \bar{r}_l + \gamma \sum_{s'\in S} P(s'|s,a,\alpha_l^*)V_{\pi,l+1}(s'|\alpha_{l+1}^*) \tag{12}$$

where $\bar{r}_l = \sum_{s'\in S} P(s'|s,a,\alpha_l^*)r_l$, and $a = \pi(s,l)$.

Similarly, we also define the $Q$-function as:

$$Q_{\pi,l}(s,a|\alpha_l^*) = \bar{r}_l + \gamma \sum_{s'\in S} P(s'|s,a,\alpha_l^*)V_{\pi,l+1}(s'|\alpha_{l+1}^*) \tag{13}$$

We further consider the agents are strategic and hence care only about individual rewards. The goal of each agent is to find an optimal oblivious policy $\pi^*$, such that,

$$V_{\pi^*,l}(s|\alpha_l^*) \geq V_{\pi,l}(s|\alpha_l^*) \ \forall s \in \mathcal{S}, \ \forall l \in [\tau]. \tag{14}$$

Let $\alpha^* = [\alpha_0^*, \cdots, \alpha_{\tau-1}^*] \in [0,1]^{\tau \times |\mathcal{A}|}$, then we can define the optimal oblivious strategy:

**Definition 2.** *The set $\mathcal{P}(\alpha)$ is the set of the optimal oblivious strategies which are chosen from the $Q$-function generated by $\alpha$. In other words, for a given $\alpha$, a policy $\bar{\pi} \in \mathcal{P}(\alpha)$ if and only if*

$$\bar{\pi}(s,l) \in \arg\max_a Q_{\bar{\pi},l}(s,a|\alpha_l) \forall s \in S \ l \in [\tau] \tag{15}$$

Here, the policy $\bar{\pi}(s,l)$ is used at $l$th time index in an episode so that the $Q$-value $Q_{\pi,l}(s)$ is maximized for all states $s \in \mathcal{S}$. *Note that $\bar{\pi}(s,l)$ does not depend on the distribution $\alpha^*$ explicitly. Hence, it is an oblivious strategy where each agent takes its decision based on its own observed state only.* Since the reward function is bounded and $\gamma < 1$, the set $\mathcal{P}(\alpha)$ is always non-empty. However, finding the optimal action is challenging for an oblivious strategy profile. We denote the initial population state distribution denoted by $f_0$. We note that as $\alpha_t$ evolves, the population state distribution $f_t$ also evolves. After convergence, for a time index $j$ in any episodes, the population state distribution will converge to the limiting population state distribution $f_j^*$, or

$$||f_{k\tau+j} - f_j^*||_2 \to 0. \tag{16}$$

### 2.4. Stationary Mean-Field Equilibrium

Throughout this paper, we seek to compute a mean-field equilibrium and action strategy. Thus, the action was taken by an agent only depends on its state independent of its episode. Further, such an action profile should converge to a stationary action distribution and state distribution. We now formally define a mean-field equilibrium.

**Definition 3** ([24]). *We say that Mean-Field Equilibrium (MFE) is achieved by an oblivious strategy $\bar{\pi}(\cdot)$, if the strategy for the players, population action, and the state distribution is such that:*

- *Each player i optimizes its expected discounted payoff assuming that population action distribution $\alpha$ is fixed; i.e., it satisfies (15).*
- *For strategy $\bar{\pi}$ of any player i, the fixed population action distribution $\alpha$ satisfies*

$$\alpha_j(a) = \frac{1}{n}\sum_s \mathbb{1}_{\bar{\pi}(s',j)=a} p(s'|s,a,\alpha_{j-1}) \ \forall j \in [\tau] \tag{17}$$

　　*We define the above as $\alpha \in \hat{D}(\bar{\pi}, f)$*
- *For $\alpha$ and $\bar{\pi}$, the state distribution $f$ satisfies*

$$f(s'_j) = \sum_s f(s) p(s'|s, \pi(s), \alpha_{j-1}) \ \forall j \in [\tau] \tag{18}$$

*which we denote as $f \in D(\bar{\pi}, \alpha)$.*

Specifically, if we fix $\alpha$ and each agent takes action $\bar{\pi}$ which belongs to $\mathcal{P}(\alpha)$, then the action distribution should return to $\alpha$ as it is invariant under the transition probability. Further, if we fix $\alpha$ and each agent takes action $\bar{\pi}$, the state distribution should give back $f$.

Since the players are learning an oblivious strategy, no agent observes the states or actions of the other agents. Also, *an agent does not know the probability transition matrix, and reward function* and will try to estimate it from the past observations as described in the next section.

## 3. Proposed Algorithm

In this section, we propose an algorithm, which will be shown to converge to the mean-field equilibrium (MFE) in the following section. For each agent $i$, the algorithm begins with a prior distribution $g$ over the stochastic game with state-space set $S$ and action space set $A$ and time horizon $\tau$. The prior distribution $g$ for modeling state transition probability distribution is typically taken to be Dirichlet distribution [14,15].

The game is played episodes $k = 0, 1, 2, \cdots$. The length of each episode is given by $\tau$. In each episode, the game is played in discrete steps, $j = 0, 1, \cdots, \tau - 1$. The episodes begin at times $t_k = k\tau, k = 0, 1, 2, \cdots$. At each time $t = k\tau + j$, the state of the agent is given by $s_t$, it selects an action $a_t$, and observes a scalar reward $r_t$ then transitions to the state $s_{t+1}$. Let $H_t = (s_1, a_1, r_1, \ldots s_t, a_{t-1}, r_{t-1})$ denote the history of the agent till time $t$.

The proposed algorithm is described in Algorithm 1. At the beginning of each episode, the MDP, $\mathcal{M}_k$ is sampled from the posterior distribution conditioned on the history $H_{t_k}$ in Line 4. We note that the sampling of MDP only relates to the sampling of the transition probability $P$ and the reward distribution since the rest of the parameters are known. We note that the algorithm doesn't perform explicit exploration like an $\epsilon$-greedy algorithm. Instead, the algorithm samples a new MDP $\mathcal{M}_k$ for episode $k$ in Line 4. The Algorithm can generate a new trajectory from the new policy [14,15] solved for the sampled MDP $\mathcal{M}_k$. We assume that after some samples, $\alpha_k$ has converged. The proposed algorithm converges as the induced transition probability and reward function converge after $\alpha_k$ converge.

---

**Algorithm 1** Proposed Algorithm for Mean-Field Game with Best Response Learning Dynamics.

---

1: **Input:** Prior distribution $g$, time horizon $\tau$, $\gamma$
2: Initialize $H_0 = \phi$.
3: **for** episodes $k = 0, 1, 2, \cdots$ **do**
4:　　Sample $\mathcal{M}_k \sim g(\cdot|H_{k\tau})$ .
5:　　Obtain optimal $Q$ for $\mathcal{M}_k$ from Algorithm 2
6:　　**for** time steps $j = 0, \ldots, \tau - 1$ **do**
7:　　　　Play $a_j = \arg\max_a Q_j(s_j, a)$.
8:　　　　Observe reward $r_j$, action of the agent $a_j$, and next state $s_{j+1}$.
9:　　　　Append action taken $a_j$, reward obtained $r_j$, and state update $s_{j+1}$ to history

$$H_{k\tau+j+1} = H_{k\tau+j} \cup \{a_j, r_j, s_{j+1}\}.$$

10:　　**end for**
11: **end for**

---

We use Backwards Induction algorithm [25] described in Algorithm 2 to obtain the Q-value function for the current sampled MDP (Line 5, Algorithm 1). Backward induction in Algorithm 2 starts from the end of the episode and calculates the potential maximum

rewards for each state and action (Line 5). The algorithm then goes back in the episode (Line 8), to calculate the maximum possible cumulative rewards for each state and action in Line 11. After all the time indices in an episode are covered, the algorithm returns the calculated optimal Q-values. We obtain the policy $\pi_k$ from the calculated Q-values and the policy is not altered in an episode. Recall for a given $\alpha$, a policy $\pi \in \mathcal{P}(\alpha)$ if and only if $\pi_k(s, j) \in \arg\max_a Q_{\pi_k, j}(s, a | \alpha_j)$ for all $s \in \mathcal{S}$ and $j = 0, 1, \ldots, \tau - 1$. Let $\alpha_k$ be the population action distribution in episode $k$, then the algorithm aims to choose a policy $\pi_k \in \mathcal{P}(\alpha_k)$. In order to choose the policy $\pi_k$ from the set $\mathcal{P}(\alpha_k)$, we use lower myopic learning dynamics, where at each episode we choose the strategy which is the smallest action index in the set $\mathcal{P}(\alpha_k)$.

---

**Algorithm 2** Backwards Induction Algorithm.

---

1: **Input:** $\mathcal{M} = \{\mathcal{S}, \mathcal{A}, P, r, \tau, \gamma\}$        $\triangleright$ Sampled MDP from Algorithm 1
2: Initialize $Q_l(s, a) = 0 \; \forall \, s \in \mathcal{S}, a \in \mathcal{A}, l \in [\tau]$.
3: **for** state $s \in \mathcal{S}$ **do**
4:      **for** state $a \in \mathcal{A}$ **do**
5:          Update Q-value function for last action

$$Q_{\tau-1}(s, a) = \sum_{s' \in \mathcal{S}} P(s'|s, a) r(s, a, s')$$

6:      **end for**
7: **end for**
8: **for** time steps $l = \tau - 2, \cdots, 0$ **do**
9:      **for** state $s \in \mathcal{S}$ **do**
10:        **for** state $a \in \mathcal{A}$ **do**
11:           Update Q-value function

$$Q_l(s, a) = \sum_{s' \in \mathcal{S}} P(s'|s, a) \times$$
$$\left( r(s, a, s') + \gamma \arg\max_a Q_{l+1}(s', a) \right) \tag{19}$$

12:        **end for**
13:      **end for**
14: **end for**
15: **Return:** $Q_l(s, a) \; \forall \, l, s, a$

---

We note that $\mathcal{M}_k$ is used in the algorithm instead of $\mathcal{M}^*$ where $\mathcal{M}^*$, the true distribution, is not known. In order to obtain an estimate, each agent samples a transition probability matrix according to the posterior distribution. Each agent follows the strategy $\pi_k$ according to the Q-values over the episode. Based on the action decision by each agent, we update the value function and the Q-function based on the obtained reward functions which depend on the value of $\alpha_k$. The detailed algorithm steps can be seen in Algorithm 1. We note that, as the algorithm converges, the value of $\alpha$ converges, and thus all the transition probabilities and value functions depend on the limiting distribution.

## 4. Convergence Result

In this section, we'll show that if the oblivious strategy is chosen according to the proposed algorithm, then the oblivious strategy $\pi$ and the limiting population action distribution $\alpha$ constitutes a Mean-Field Equilibrium (MFE). More formally, we have obtained the following–

**Theorem 1.** *The optimal oblivious strategy obtained from Algorithm 1 and the limiting action distribution constitute a mean-field equilibrium and the value function obtained from the algorithm converges to the optimal value function of the true distribution.*

The rest of the section proves this result. We first note that the lower-myopic best response strategy leads to a convergence of the action strategy following the results in [10] for finite action space and state space. We note that there might be multiple actions that can maximize the state-action value function. This may lead to choosing different actions at different iterations for the same state. To avoid the oscillations between the best actions and hence keep the policy stable, we choose a lower-myopic strategy. This lower-myopic strategy avoids conflicts when the agents have a non-unique strategy that maximizes the value function. Further, any way of resolving the multiple optima could be used, including upper-myopic giving the same result. Having shown that $\alpha$ converges, we now proceed to show that the converged point of the algorithm results in an MFE.

We first show the conditions needed for a policy $\pi$, a population state $f$, and action distribution $\alpha$ to constitute an MFE (Section 4.1). Then, we show that the conditions for the policy to be MFE given in Section 4.1 are met for any optimal oblivious strategy (Section 4.2). Thus, the key property that is required to show the desired result is that the proposed algorithm leads to an optimal oblivious strategy. In order to show that, we show that the value function of the sampled distribution converges to the true distribution (Section 4.3). The result in Section 4.3 shows that the value function iterates eventually converge to the value function with knowledge of the true underlying distribution of the transition probability $\mathcal{M}^*$, thus proving that the proposed algorithm converges to an optimal oblivious strategy which constitutes a mean-field equilibrium thus proving the theorem.

### 4.1. Conditions for a Strategy to Be a MFE

In this section, we will describe the conditions for an oblivious strategy $\pi$ to be a MFE. Recall that, in Section 2.2, we defined two maps $\mathcal{P}(\alpha)$ and $\mathcal{D}(\pi, \alpha)$. For a given action coupled stochastic game, the map $\mathcal{P}(\alpha)$ for a given population action distribution $\alpha$ gives the set of the optimal oblivious strategies. Further, the map $\mathcal{D}(\pi, \alpha)$ for a given population action distribution $\alpha$ and oblivious strategy $\pi$ gives the set of invariant population state distribution $f$.

We define the map $\hat{\mathcal{D}}(\pi, f)$ which gives the induced population action distribution $\alpha$ induced from the oblivious strategy $\pi$ and the population state distribution $f$. The following lemma gives the conditions that the stochastic game constitutes a mean-field equilibrium. These conditions have been provided in [11], and the reader is referred to [11] for further details and proof of this result.

**Lemma 1** (Definition 7 [11]). *An action coupled stochastic game with the strategy $\pi$, population state distribution $f$ and population action distribution $\alpha$ constitute a mean-field equilibrium if $\pi \in \mathcal{P}(\alpha), f \in \mathcal{D}(\pi, \alpha)$ and $\alpha \in \hat{\mathcal{D}}(\pi, f)$.*

### 4.2. Conditions of Lemma 1 Are Met for Any Optimal Oblivious Strategy

In this section, we show that the conditions of Lemma 1 are met for any optimal oblivious strategy. In the mean-field equilibrium, each agent plays according to the strategy $\pi \in \mathcal{P}(\alpha)$. If the average population action distribution is $\alpha$, and each agent takes an oblivious strategy, hence, we must have the evolution of the state space such that the oblivious strategy on those states leads to an average action distribution of $\alpha$. Since we assume a large number of agents, including agent $i$'s own state will not change the population state distribution. So, we let the average population state distribution be $f_j$ at time index $j$ as,

$$f_j(s) = \frac{1}{n} \sum_{m \in [n]} \mathbb{1}_{s_{m,j}=s} \tag{20}$$

where $s_{m,j}$ is the state of the agent $m$ at time index $j$. Similarly, we also include agent $i$'s action as well in the population action distribution $\alpha$. Then, we have,

$$\mathbf{E}[f_j(s)] = \mathbf{E}\left[\frac{1}{n}\sum_{m\in[n]}\mathbb{1}_{s_{m,j}=s}\right] \tag{21}$$

$$= \frac{1}{n}\sum_{m\in[n]}\mathbf{E}\left[\mathbb{1}_{s_{m,j}=s}\right] \tag{22}$$

$$= \frac{1}{n}\sum_{m\in[n]}P\big(\{s_{m,j}=s\}\big) \tag{23}$$

$$= \frac{1}{n}\sum_{m\in[n]}\sum_{s'\in\mathcal{S}}P\big(\{s_{m,j-1}=s\}\big)P\big(s|s',\pi(s'),\alpha_{j-1}\big) \tag{24}$$

$$= \sum_{s'\in\mathcal{S}}\left(\frac{1}{n}\sum_{m\in[n]}P\big(\{s_{m,j-1}=s\}\big)\right)P\big(s|s',\pi(s'),\alpha_{j-1}\big) \tag{25}$$

$$= \sum_{s'\in\mathcal{S}}\mathbf{E}\big[f_{j-1}(s')\big]P\big(s|s',\pi(s'),\alpha_{j-1}\big) \tag{26}$$

where $P(\{s_{m,j} = s\})$ is probability of agent $m$ being in state $s$ at time index $j$. Here, Equation (24) follows from the transition probability matrix. Recursively replacing $f_{j-1}$ in Equation (26) using Equation (21) for all $j \in [\tau]$ gives the required result of $f \in \mathcal{D}(\pi, \alpha)$.

The above statement also implies that $\alpha$ must satisfy

$$\alpha_j(a) = \sum_{\pi^{-1}(a,j)} f_j(s) \tag{27}$$

where $\pi^{-1}(a, j)$ represents the set of states $s$ for which $a \in \pi(s, j)$. This is equivalent to saying that if all the agents follow the optimal oblivious strategy $\pi \in \mathcal{P}(\alpha)$, then the population state distribution $f$ and the population action distribution $\alpha$ satisfy $f \in \mathcal{D}(\pi, \alpha)$ and $\alpha \in \hat{\mathcal{D}}(\pi, f)$.

### 4.3. Sampling Does Not Lead to a Gap for Expected Value Function

In the last subsection, we proved that there exists an optimal oblivious strategy. We will show that the policies generated by Algorithm 2, $\pi_k$, for the sampled system dynamics $\mathcal{M}_k$ in episode $k$ by Algorithm 1 converges to the optimal oblivious policy $\bar{\pi}$. To show the convergence of the policy $\pi_k$ to $\bar{\pi}$, we will show that the value function of the optimal oblivious policy $\pi_k$ converges to the optimal value function of the true system dynamics.

We will first describe the lemmas that are used to prove the required result. We start by stating the Azuma-Hoeffding Lemma for obtaining confidence intervals.

**Lemma 2** (Azuma-Hoeffding Lemma [15]). *If $Y_n$ is a zero-mean martingale with almost surely bounded increments, $|Y_i - Y_{i-1}| \le C$, then for any $\delta \ge 0$ with probability at least $1 - \delta$, $Y_n \le C\sqrt{2n\log(1/\delta)}$.*

We also utilize the following result of [15] on any $\sigma$-measurable function $g$.

**Lemma 3.** *If $f$ is the distribution of $\mathcal{M}^*$ then, for any $\sigma(H_{t_k})$-measurable function $g$,*

$$\mathbf{E}[g(\mathcal{M}^*)|H_{t_k}] = \mathbf{E}[g(\mathcal{M}_k)|H_{t_k}]. \tag{28}$$

At the start of every episode, each agent samples system dynamics from the posterior distribution given $H_{t_k}$. The following result bounds the difference between the optimal value function learned by the true distribution $\mathcal{M}^*$ function using the optimal policy $\pi^*$

which is unknown, and the optimal value function achieved by the sampled distribution $\mathcal{M}_k$ from the policy $\pi_k$.

**Lemma 4.** *Let* $V^{\mathcal{M}_k}_{\pi_{k,j}}(s|\alpha_j))$ *be the optimal value function for an oblivious policy* $\pi_k(s,j) = \arg\max\limits_a Q^{\mathcal{M}_k}_{\pi_{k,j}}(s,a|\alpha_l)$ *for the sampled system dynamics* $\mathcal{M}_k$ *chosen form Algorithm 1. Then* $V^{\mathcal{M}_k}_{\pi_{k,j}}(s|\alpha_j))$ *converges to the optimal value function,* $V_{\pi^*,j}(s|\alpha_j)$, *of the true system dynamics* $\mathcal{M}^*$ *i.e., for all states* $s \in \mathcal{S}$ *as* $k \to \infty$ *with probability at least* $1 - \delta$,

$$V^{\mathcal{M}_k}_{\pi_{k,j}}(s|\alpha_j)) - V_{\pi^*,j}(s|\alpha_j) \to 0 \tag{29}$$

**Proof.** We note that since the optimal value function is a $\sigma(H_{t_k})$-measurable, we can use Lemma 3 to bound the difference of the optimal value functions of the sampled distribution at episode $k$ and the true distribution to show that for all states $s \in \mathcal{S}$,

$$\mathbf{E}\left[V_{\pi^*,j}(s|\alpha_j) - V^{\mathcal{M}_k}_{\pi_{k,j}}(s|\alpha_j))\right] = 0 \tag{30}$$

Note that the length of all episodes is given by $\tau$ and the support of the reward is [0, 1]. Therefore for all states $s \in \mathcal{S}$, we have $V_{\pi^*,0}(s|\alpha_0) - V^{\mathcal{M}_k}_{\pi_{k,0}}(s|\alpha_0)) \in [-\tau, \tau]$. Note that this condition is similar to bounded increments in Azuma-Hoeffding Lemma (Lemma 2).

Since $V_{\pi^*,0}(s|\alpha_0) - V^{\mathcal{M}_k}_{\pi_{k,0}}(s|\alpha_0) \in [-\tau, \tau]$ is a zero mean martingale with respect to the filtration $\{H_{t_k} : k = 1, .., m\}$, and satisfies the assumptions of Azuma-Hoeffding Lemma, we obtain the result as in the statement of the Lemma. Also, for all states $s \in \mathcal{S}$, we have, $V_{\pi^*,0}(s|\alpha_0) - V^{\mathcal{M}_k}_{\pi_{k,0}}(s|\alpha_0)) \in [-\tau, \tau]$. So, the difference is a zero-mean martingale and has the bounded increments property. Applying the Azuma-Hoeffding Lemma to the martingale, we have the following result,

$$\sum_{k=1}^{m}\left(V_{\pi^*,0}(s|\alpha_0) - V^{\mathcal{M}_k}_{\pi_{k,0}}(s|\alpha_0)\right) \le \tau\sqrt{2m\log(1/\delta)} \tag{31}$$

For total time $T$ of the algorithm, we have $m = T/\tau$. Thus, we obtain,

$$\sum_{k=1}^{m}\left(V_{\pi^*,0}(s|\alpha_0) - V^{\mathcal{M}_k}_{\pi_{k,0}}(s|\alpha_0)\right) \le \tau\sqrt{2\frac{T}{\tau}\log(1/\delta)} \tag{32}$$

$$= \sqrt{2T\tau\log(1/\delta)} \tag{33}$$

Thus, for all $\theta > 1/2$, as $T \to \infty$, we have

$$\frac{\sum_{k=1}^{\lceil T/\tau \rceil}\left(V_{\pi^*,0}(s|\alpha_0) - V^{\mathcal{M}_k}_{\pi_{k,0}}(s|\alpha)_0\right)}{T^\theta} \to 0 \tag{34}$$

Substituting $\theta = 1$, the above expression says that $\tau$ times the average difference in an episode which converges to zero as total time $T \to \infty$, which gives us the convergence of the optimal value functions of the two distributions. Thus, we have

$$V_{\pi^*,0}(s|\alpha_0) - V^{\mathcal{M}_k}_{\pi_{k,0}}(s|\alpha_0) \to 0 \text{ as } k \to \infty \tag{35}$$

□

## 5. Conclusions

We consider an action-coupled stochastic game consisting of a large number of agents where the transition probabilities are unknown to the agents. We utilize the concept of mean-field equilibrium where each agent's reward and the transition probability are only impacted through the mean distribution of the actions of the other agents. When the

number of agents grows large, the mean-field equilibrium becomes equivalent to the Nash equilibrium. We propose a posterior sampling-based approach where each agent draws a sample using an updated posterior distribution and selects an optimal oblivious strategy accordingly. We show that the proposed algorithm converges to the mean-field equilibrium without knowing the transition probabilities apriori.

This paper shows asymptotic convergence to the mean-field equilibrium while finding the convergence rate is an interesting future direction. Further, the convergence rate in the number of users to the mean-field limit (akin to [26], while for the mean-field game rather than mean-field control) is an important direction.

**Author Contributions:** All the authors contributed significantly to the work. All authors have read and agreed to the published version of the manuscript.

**Funding:** This research was supported in part by the National Science Foundation under grants CCF-1527486 and CNS-1618335.

**Institutional Review Board Statement:** Not applicable.

**Informed Consent Statement:** Not applicable.

**Conflicts of Interest:** The authors declare no conflict of interest.

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
