# Peer review of "Reinforcement Learning for Mean-Field Game"

_algorithms, doi:10.3390/a15030073_

Round 1

Reviewer 1 Report

Though the paper is mathematically sound, I have several concerns regarding its clarity and contribution significance.

As to clarity, I mention the following two major problems:

  • the introduction does not follow a clear progression from the description of the general context to the definition of the research gap and the problem to the proposal of solution and the claims about results and contribution. In particular, the authors first introduce the context of multiple-agents games and then focus on mean-field games, providing examples of such games. However, soon afterwards they mix assumptions and results (lines 59-68) and then go back to a very brief literature review (lines 69-77) to end the Introduction with a mixture of references to the past literature to highlight the different assumptions (which should be done earlier) and brief statements of contribution.
  • the roadmap the authors follow in each section is never sketched at its beginning, which would help the reader follow the authors' reasoning. Also, many steps would gain clarity from a description in plain English before the mathematical formulation. This is the case, e.g., for the Value function and the Q function   

The original contribution and its difference from the past literature are not stated quite clearly. The authors first state that they adopt a mean-field approach where the Q function of an agent is affected by the population choices (lines 35-36) but then they state to adopt an oblivious strategy, where considering other agents' actions does not matter to achieve the best performance (lines 135-138) and devote some space to prove this result or its generalization (Section 4.2). But the major contribution of the paper seems to be the proposal of the algorithm in Section 3 that allows finding the mean-field game solution (again) when transition probabilities are unknown and must be estimated. At the end, the reader is a bit puzzled. Please clearly state the assumptions and stick with them.

If the original contribution is the algorithm in Section 3, some results about the statistical properties of the estimate should be provided. And also some results about the convergence speed since players may not have an infinite time. As such, the contribution is not huge, and the authors should make an effort to compare it against the present literature (in one place, please, rather than scattered throughout the paper) and prove its significance.

Also, please note that there are several dangling sentences. For example, in lin213, the sentence "More formally, we have"...(what?)

Finally, the use of the results extracted from Ref. [6] is a bit easy-going. Please state clearly what is proven in [6] already and what you obtain further, or where you step away from [6]. Lemma 1 is not proven (is it taken fully from [6]?). Section 4.2 should probably accompany Lemma 1 rather than constituting a section apart.

Author Response

Though the paper is mathematically sound, I have several concerns regarding its clarity and contribution significance.

As to clarity, I mention the following two major problems:

  • the introduction does not follow a clear progression from the description of the general context to the definition of the research gap and the problem to the proposal of solution and the claims about results and contribution. In particular, the authors first introduce the context of multiple-agents games and then focus on mean-field games, providing examples of such games. However, soon afterwards they mix assumptions and results (lines 59-68) and then go back to a very brief literature review (lines 69-77) to end the Introduction with a mixture of references to the past literature to highlight the different assumptions (which should be done earlier) and brief statements of contribution.
  • the roadmap the authors follow in each section is never sketched at its beginning, which would help the reader follow the authors' reasoning. Also, many steps would gain clarity from a description in plain English before the mathematical formulation. This is the case, e.g., for the Value function and the Q function   

Answer: We have rewritten the Introduction.  We have now divided Introduction in three subsections Motivation, Contributions, and Related Literature.

The original contribution and its difference from the past literature are not stated quite clearly. The authors first state that they adopt a mean-field approach where the Q function of an agent is affected by the population choices (lines 35-36) but then they state to adopt an oblivious strategy, where considering other agents' actions does not matter to achieve the best performance (lines 135-138) and devote some space to prove this result or its generalization (Section 4.2). But the major contribution of the paper seems to be the proposal of the algorithm in Section 3 that allows finding the mean-field game solution (again) when transition probabilities are unknown and must be estimated. At the end, the reader is a bit puzzled. Please clearly state the assumptions and stick with them.

Answer: 

We seek to obtain a model-based RL algorithm to find the mean-field equilibrium in an episodic-set up. To the best of our knowledge, this is the first work on episodic RL algorithm for mean-field equilibrium. We consider an oblivious strategy [10, 11], where each agent takes an action based only on its state. Thus, even though the transition probability and reward depend on the empirical distribution of the agents' action and states, an agent only seeks policy based on its own state. Hence, an agent does not need to track the policy state of the other agents.   

In our algorithm, we maintain a history of the samples, and sample a MDP which fits the history with a high confidence at the start of the episode. The policy is then computed with this MDP. The agents then play the computed policy and collect samples over the steps of the episode. We show that such an algorithm converges to the equilibrium. 

We have added the above discussion in discussion in now clearly mentioned our Contributions in Introduction under Section 1.2. Further, to expand on the comparisons with the previous literature, we have added the following discussion in Section 1.3 of the Introduction

Unlike the standard literature on the mean-field equilibrium on stochastic games [10, 12, 13], we consider that the transition probabilities are unknown to the agents. Instead, each agent learns the underlying transition probability matrix using a readily implementable posterior sampling approach [14,15]. 

All agents employ best response dynamics to choose the best response strategy which maximizes the (discounted) payoff for the remaining episode length. We show the asymptotic convergence of the policy and the action distribution to the optimal oblivious strategy and the limiting action distribution respectively. We estimate the value function using backward induction and show that the value function estimates converge to the optimal value function of the true distribution. We also use the compactness of state and action space to show that the converged point constitutes a mean-field equilibrium (MFE).

[5] considers a variant of the Mean-field game where the state is the same across the agents. Unlike [5], we consider a generalized version of the game where the state can be different for different agents. Further, we do not consider a game where adversarial equilibrium or coordinated equilibrium is required to be present. We also do not need to track the action and the realized Q-value of other agents as was the case in [5].

Recently, the authors of [16] studied a policy-gradient based approach to achieve mean-field equilibrium. The authors of [17,18] considered variants of Q-learning  to achieve Nash equilibrium.  Actor-critic based algorithms have been analyzed in [19]. The authors of  [20] consider a deep neural network based algorithm for mean-field games. In contrast, this paper considers a posterior sampling-based approach. As noted in [21], model-based approaches converge faster than the model-free approaches in general. Thus, understanding of theoretical properties of the model-based approaches is an important problem. Further, all these papers require huge storage space as the policy depends on the actions of the other agents whereas in our setting we provide a policy which is oblivious of the states and actions of other agents. 

The authors of [22,23] compute mean-field equilibrium for a setting where the evolution of the state of an agent does not depend on the action or states of the other agents. However, in our setting, we consider a generic setting where the both reward and the next state of an agent depends on the actions as well as the states of the other agents. Thus, though the mean field game converges to the potential game in the above papers, in our setting, the game does not converge to the potential game. Hence, finding an equilibrium strategy is more challenging in our setting. The key contribution of the paper is a Posterior-sampling based algorithm that is used by each agent in a multi-agent setting, which is shown to converge to a mean-field equilibrium. The proposed algorithm does not assume the knowledge of transition probabilities and learns them using a posterior sampling approach. Further, our algorithm is computationally efficient as the policy only depends on the current state of an agent, i.e., it is oblivious of the states and actions of other agents. 

If the original contribution is the algorithm in Section 3, some results about the statistical properties of the estimate should be provided. And also some results about the convergence speed since players may not have an infinite time. As such, the contribution is not huge, and the authors should make an effort to compare it against the present literature (in one place, please, rather than scattered throughout the paper) and prove its significance.

Answer: We have now clearly explained ourexplained what our contribution in Section 1.2 of the Introductions.

Also, please note that there are several dangling sentences. For example, in lin213, the sentence "More formally, we have"...(what?)

Answer: We have now fixed the typo (Section 4, page 8)

Finally, the use of the results extracted from Ref. [6] is a bit easy-going. Please state clearly what is proven in [6] already and what you obtain further, or where you step away from [6]. Lemma 1 is not proven (is it taken fully from [6]?). Section 4.2 should probably accompany Lemma 1 rather than constituting a section apart.

Answer: We have added the corresponding result from the arXiv version of Adlakha and Johari. We keep this Lemma separated from Section 4.2 for ease in reading. We mention this reasoning in line 242 in the paragraph above Section 4.1.

Reviewer 2 Report

I have two comments for this work:

1- The contribution is great, but the articulation of the content is not good. Please benefit from some diagrams to improve the readability of the paper.

2- Please use big oh and big omega notation to demonstrate the cost of the RM learning in the best and worst case  

Author Response

: Thanks a lot for appreciating the work. We note that the paper is theoretical, and the system model of stochastic game is well explained in the manuscript. Thus, we believe that the diagrams will not be able to better present the mathematical system model for stochastic games. Also, we do not have any cost of RM learning in the paper, so we are not sure as to which results the reviewer is mentioning to add the order notations. In general, we believe that precise results are better than the order results, and the pre-constants may hide some factors of importance to the application.

Round 2

Reviewer 1 Report

The authors have complied with all my comments.